# Associations between Psychological Variables, Knowledge, Attitudes, Risk Perceptions and Health Behaviours towards COVID-19 among Adolescents

**DOI:** 10.3390/jcm11164793

**Published:** 2022-08-16

**Authors:** Ángela Asensio-Martínez, Alejandra Aguilar-Latorre, Olga García-Sanz, Bárbara Oliván-Blázquez, Yolanda López-del-Hoyo, Rosa Magallón-Botaya

**Affiliations:** 1Institute for Health Research Aragón (IIS Aragón), 50009 Zaragoza, Spain; 2Research Network on Chronicity, Primary Care and Health Promotion (RICAPPS, RD21/0016/0005), Carlos III Health Institute, 28029 Madrid, Spain; 3Department of Psychology and Sociology, University of Zaragoza, 50009 Zaragoza, Spain; 4Institute of Secondary Education San Miguel (Ministry of Education, Universities, Culture and Sports of the Government of the Canary Islands), 38618 Santa Cruz de Tenerife, Spain; 5Department of Medicine, Psychiatry and Dermatology, University of Zaragoza, 50009 Zaragoza, Spain

**Keywords:** adolescents, COVID-19, health behaviours, risk perceptions, psychological variables

## Abstract

There is currently little scientific evidence available that allows us to understand patterns of knowledge, risk perception, attitudes, and behaviours among adolescents in relation to COVID-19. This study aims to analyse the relationship between knowledge about COVID-19, risk perception, and psychological variables and the adherence to preventive measures among the adolescent population. It is a descriptive cross-sectional study, which included adolescents between the ages of 12 and 18 (*n* = 354). The questionnaire was sent to several secondary schools chosen by convenience sampling and following a non-probabilistic snowball sampling. Descriptive, univariate, and multivariate analyses were carried out in order to determine whether knowledge about COVID-19, risk perception, tolerance of frustration, planning and decision-making, family functionality, self-efficacy, self-esteem, and social skills are related to preventive measures. The adoption among adolescents of behaviours which protect them against COVID-19 depends on knowledge about the disease, the perception of the risk it poses to them, as well as their tolerance of frustration and planning and decision-making abilities. The relationship between the individual variables among adolescents with the adoption of behaviours which protect them against COVID-19 has been confirmed. The development of intervention and communication strategies that take the psychosocial situation of adolescents into account will help to increase the adoption of protective health behaviours in the context of a pandemic.

## 1. Introduction

The global situation experienced since March 2020 due to the COVID-19 (SARS-CoV-2) pandemic makes it essential for governments to determine the variables that influence compliance with the preventive measures established to combat the current pandemic. The World Health Organisation (WHO) itself has warned that the best way to prevent and slow transmission is to be well informed about the COVID-19 virus, the disease it causes and how it spreads [1].

However, there is currently little scientific evidence available to help us to understand patterns of knowledge, perception, attitude and behaviour among the population [2]. Surveys on knowledge, attitudes and behaviours related to COVID-19 are useful for managing different communication strategies where there is a lack of knowledge, or for supporting chosen strategies with fresh knowledge. Gathering data on these factors is important for the authorities, as it allows them to strengthen established communication strategies and modify others whose indicators show room for improvement [3].

Some research on adolescents has linked a social skills deficit to the use of substances that are harmful to health such as alcohol, tobacco, or cannabis [4,5]. In terms of self-esteem, some studies found no significant associations between self-esteem and the consumption of illicit substances or alcohol, while others yielded results in which self-esteem was significantly associated with at least one risk behaviour among adolescents [6,7].

In general, in the studies that relate self-efficacy to addictive behaviour in young people, there is a strong consensus among authors in considering it as a protective factor against the time of onset of consumption, maintenance behaviour, and possible relapses [8,9,10]. However, some results obtained in more recent studies do not corroborate this relationship between self-efficacy and drug use, but find that efficacy is a broader construct than typically considered in prevention [11,12].

Low frustration tolerance has been linked to drug use as it is a quick means of achieving immediate pleasure for individuals with little ability to delay gratification and satisfaction of their needs [7].

The family, as an agent of socialisation, has also been strongly related to preventive and risk behaviours in the field of adolescent health. Numerous studies have found that family variables such as attitude towards consumption, the family bond, permissiveness in parental style, family drug use, family composition, patterns of family interaction, and discrepancies in family perceptions are closely related to adolescent drug consumption [13,14,15,16,17].

Public risk perception is an effective strategy for governments seeking to encourage the public to adopt recommendations for protective action during the COVID-19 pandemic. This variable was significantly associated with public adoption of recommendations for protective action [18]. 

Health behaviours and risk perception are concepts widely used in public health. Health behaviours are practises made by persons that impact health or mortality and can either benefit or harm the actor’s or others’ health [19]. Risk perception is linked to prevention and protection behaviours against various diseases or events, and its implications for risk communication [20,21]. It has been shown that risk perception can influence health-related behaviours and change risk behaviours [22]. Several studies on the adolescent population relate risk perception to substance use [7,13,23,24,25], which underlines the importance of risk perception in the initiation and maintenance of risky health behaviours.

In terms of protective health behaviours related to COVID-19 (compliance with the preventive measures set by the health authorities, e.g., safety distance, hand washing, wearing a mask) or risk behaviours (non-compliance with these measures), almost total compliance with the use of masks and hand washing was found. Differences in attendance at gatherings were observed between age groups. Participation in social gatherings increased the younger the age group [3].

Preventive measures incorporate behavioural measures [26]. Therefore, it is of vital importance to know the influence that different variables exert on compliance with infection-preventing behaviours [27,28,29]. This kind of research on the under-18 age group is very scarce, even though it is a population group that has had a large number of infections across the different waves and has transmitted the virus to other population groups.

The present study attempts to shed light on the relationship that knowledge about COVID-19, risk perception, and psychological variables (such as tolerance to frustration, planning and decision-making, family functionality, self-efficacy, self-esteem, and social skills) have on adherence to preventive measures among the adolescent population.

## 2. Materials and Methods

### 2.1. Design

Cross-sectional study of Spanish adolescents.

### 2.2. Participants and Sample Size

The research was carried out on a representative sample of the Spanish adolescent population aged 12 to 18 years. According to the Spanish National Statistical Institute (INE), there are a total of 3,504,061 Spanish residents aged between 12 and 18 [30]. With a margin of error of 5%, a probability of success of 95%, a confidence level of 95%, a precision level of 3%, and an estimated withdrawal rate of 20%, the sample size required was approximately 253 adolescents. The sample size finally achieved was 354, comfortably exceeding this requirement. Calculations were performed with the free Epidat software, version 4.2 [31].

The questionnaire was sent to several high schools, mainly situated in the autonomous regions of Zaragoza and the Canary Islands (Spain). The high schools were chosen by convenience sampling (i.e., the high schools to which we had easy access and where we were able to present the study first were chosen) [32]. The sampling technique was non-probabilistic snowball sampling (i.e., research participants were asked to encourage others to answer) [33]. The questionnaire was also disseminated via social networks (Facebook (https://www.facebook.com accessed on 1 October 2020), Instagram (http://instragram.com accessed on 1 October 2020) and Twitter (https://twitter.com accessed on 1 October 2020)).

### 2.3. Instruments

We collected sociodemographic variables (gender, place of residence, level of education, and financial aid) through an ad hoc questionnaire.

Knowledge, attitudes, risk perception, and behaviours among Spanish adolescents in relation to COVID-19 were measured using the questionnaire entitled ‘Knowledge, Attitudes, Risk Perception, and Practices among Spanish Adolescents in relation to the COVID-19 Pandemic’ [34]. This questionnaire is an adaptation of the questionnaire developed by Honarvar et al. [35]. Of the total questions, 17 score knowledge, 18 score practices, and 10 score attitudes and risk perception. The Spanish version of the questionnaire was found to have an acceptable internal consistency (ordinal alpha = 77%) [34].

The level of functioning of the family unit was measured using the Family APGAR score [36]. This shows how family members perceive the level of functioning of the family unit as a whole. It consists of 5 items that are scored between 0 and 4. It is validated for use on children and adolescents in Spain and its correlation index is between 0.71 and 0.83 [37]. The internal consistency in our sample was acceptable (α = 0.78).

Self-efficacy was measured using the Generalised Self-Efficacy Scale. This scale is based on the concept of self-efficacy expectation defined by Albert Bandura, it is used to assess the self-efficacy of adolescents regarding their own capacity. The interest of this scale lies in obtaining information on how an adolescent acts in the face of a difficulty or problem [38]. It consists of 10 items that are scored between 0 and 4 and is validated for use in Spain for participants aged 12 and over. Its Cronbach’s alpha is 0.82 [39] and the internal consistency in our sample was excellent (α = 0.90).

Self-esteem was measured using the Rosenberg self-esteem Scale. Rosenberg developed this scale in 1965 [40], considering that if one judges oneself in positive terms with acceptance and condescension, one will have a positive self-esteem, but if one evaluates oneself and gives little or no importance to one’s own characteristics, then one will lack some basic pillars for psychological survival, lacking the minimum security, esteem and self-respect that is necessary. It consists of 10 items that are scored between 1 and 4. It is validated for use in Spain for participants aged 12 and over and its Cronbach’s alpha is 0.82 [41]. The internal consistency in our sample was good (α = 0.86).

Planning and decision making were measured using the Spanish adaptation of the decision making/problem solving subscale of the Life–skills Development Scale [42]. This scale evaluates the adolescents’ perception of their own ability to plan and make decisions, and their own capacity for autonomy. It consists of 8 items that are scored between 1 and 7 and is validated for use with Spanish adolescents. Its Cronbach’s alpha is 0.89 [41] and the internal consistency in our sample was excellent (α = 0.93).

Social skills were measured using the Social Skills Scale. This scale evaluates the adolescents’ perception of their own social skills, which are cognitive or behavioural routines that allow us to maintain good relationships with others [41]. The scale consists of 12 items that are scored between 1 and 7; it is validated for use with Spanish adolescents and its Cronbach’s alpha is 0.69 [41]. The internal consistency in our sample was almost acceptable (α = 0.67).

Tolerance of frustration was measured using the Spanish adaptation of the stress management subscale of the Emotional Quotient Inventory [43]. This inventory evaluates children and adolescents’ perception of their own ability to manage stress/frustration and impulsive control. It consists of 8 items that are scored between 1 and 5; it is validated for use with Spanish adolescents and its Cronbach’s alpha is 0.77 [41]. The internal consistency in our sample was good (α = 0.84).

### 2.4. Procedure

The Google Forms platform was used to generate an online survey. From October 2020 until January 2021, the survey was accessible via an online link. The survey was sent to participants by email, social media, and was also disseminated by partner organisations and secondary schools. The data was anonymised. Adolescents under 13 years of age needed the express consent of a parent or legal guardian, who was required to confirm their date of birth before the questionnaire began.

### 2.5. Data Analysis

Firstly, a descriptive analysis was performed (frequencies and percentages for categorical variables; mean and standard deviation for continuous variables) to determine the characteristics of the sample. Secondly, to analyse the associations between protective health behaviours against COVID-19 and each of the variables, correlations were performed using the Pearson correlation coefficient test. Thirdly, a multiple linear regression was performed [44], using a stepwise method to obtain a better-fitting result upon statistical analysis. This stepwise regression simply repeats multiple regressions, deleting the least correlated variable each time [45]. Participants with missing values were eliminated following the Listwise Deletion method [46]. Data collection and statistical analyses were performed using Excel software and SPSS software (Version 25.0, IBM Corp., Armonk, NY, USA) [47]. All significant levels were established at *p* < 0.05.

## 3. Results

### Descriptive and Bivariate Analysis

Firstly, the descriptive analysis is shown in Table 1. Of the 354 adolescents, 231 were female and 119 were male. Their mean age was 15.37 ± 1.55 years. Table 1 also shows the results of the bivariate analysis of COVID-19 protective health behaviours according to sociodemographic characteristics, knowledge, attitudes, and risk perception of COVID-19, as well as psychological variables. There is a significant relationship between protective health behaviours and knowledge (0.273; *p* < 0.001), attitudes and risk perception of COVID-19 (0.335; *p* < 0.001), family APGAR (0.137, *p* = 0.010), planning and decision making (0.175; *p* = 0.001), and frustration tolerance (0.240; *p* < 0.001). Self-efficacy was not significant (0.084, *p* = 0.115), neither were self-esteem (0.070, *p* = 0.189), social skills (0.068, *p* = 0.205), nor the sociodemographic variables.

With regards to the multivariate analysis, once the stepwise regression eliminated the weakest correlated variables, the remaining variables are shown in Table 2. Attitudes and risk perceptions of COVID-19 (*b* = 0.405; *p* < 0.001), knowledge (*b* = 0.548; *p* < 0.001), frustration tolerance (*b* = 0.167; *p* = 0.003) and planning and decision making (*b* = 0.079; *p* = 0.013) are predictors of COVID-19 protective health behaviours. This model explains 19.5% of the overall variance [*R*^2^ adjusted = 0.195, *F* (4348) = 22.296, *p* < 0.001].

## 4. Discussion

This study has analysed whether various psychological variables, knowledge about COVID-19, and attitudes and risk perception predict the adoption of protective health behaviours by adolescents. The results of the study reveal that planning and decision-making, frustration, knowledge about COVID-19, and attitudes and risk perception significantly predict the adoption of protective health behaviours against COVID-19 among adolescents.

As for the relationship between knowledge about COVID-19 and the preventive measures with which the population complied, previous studies confirm how information focussed on increasing risk perception and knowledge about the disease drive the implementation of protective health behaviours [48,49]. However, such influence should be assumed with caution, because other authors report that an adequate level of knowledge does not necessarily lead to the adoption of protective health behaviours [50,51,52]. It therefore follows that more variables influence this process, a result confirmed by the present study, which has shown how, in addition to knowledge about COVID-19, the perception of the risk posed by COVID-19, tolerance of frustration, and planning and decision-making predict the adoption of protective health behaviours.

Specifically, previous studies on the risk perception from COVID-19 among adults reflect a positive impact of risk perception on the monitoring of protective health behaviours, and how risk perception mediates the association between knowledge about COVID-19 and protective health behaviours [29,35,49,53]. There are few studies of this kind that look specifically at adolescents, although the association between the perceived risk of infection and the risk behaviours in the context of the pandemic has been confirmed [54].

Likewise, studies on illness and frustration tolerance in adolescents focus mainly on the study of factors associated with addictive risk. They have confirmed that low frustration tolerance is the most common risk factor associated with addiction [55,56,57]. This data corroborates the present study, in which tolerance to frustration is one of the psychological variables that influence protective health behaviours against COVID-19. Brooks et al. (2020) [58] identified frustration and inadequate knowledge as two of the factors that would influence people’s response to the social distancing recommendation during the COVID-19 pandemic.

In terms of planning and decision-making, previous studies confirm the influence of these variables on the intention to engage in protective health behaviours [59,60].

Family functionality showed a significant correlation with protective health behaviours against COVID-19 among adolescents. However, this relationship was not statistically significant. This result may have been because for some adolescents the family may be a protective factor, but for others it may be a risk factor. The dramatic changes experienced pre-pandemic (technological developments and globalisation) exposed adolescents to a greater number of factors that influence their development, and that the elements traditionally considered as protective factors are more relative [61,62]. In addition, for some individuals, it is individual perception that determines health-promoting behaviour [63]. As for family influence on adolescents, aspects such as family behaviour models, parental control, and parenting style should be taken into account [64,65,66].

In the present study of adolescents, self-esteem and self-efficacy have not been found to have a direct influence on the adoption of protective health behaviours against COVID-19. These results corroborate previous studies in which self-esteem and self-efficacy were not found to be associated with these protective behaviours, and even presented an inverse association [11,67,68,69]. However, both self-esteem and self-efficacy have an ambiguous relationship with protective health behaviours, since other studies argue that self-esteem and self-efficacy are protective health factors among adolescents [70,71,72,73].

Likewise, social skills do not predict the adoption of protective health behaviours against COVID-19, which could be because, during the pandemic, adolescents’ forms of communication have been distorted, eliminating the possibility of developing social behaviour appropriate to their age and reducing opportunities to project assertively through their social skills [74]. Although previous studies on adolescent addiction have confirmed the relationship between social skills and protective health behaviours [75,76], the characteristics of the COVID-19 pandemic have not produced the same relationship.

In terms of the limitations of this study, a relevant question to add could be if adolescents lived with someone who was vulnerable to COVID-19. As for the family variable, other relevant variables should be considered in future research (i.e., models of family behaviour, parental control, and parenting style). It would also be interesting to analyze the facilitators and barriers that influence the modification of protective health behaviors of adolescents, through qualitative methods such as Online Photovoice (OPV), which gives opportunities to the participants to express their own experience with the least amount of manipulation possible [77].

Furthermore, future studies with larger sample size could perform subgroup analyzes (i.e., divide the sample according to whether they have high or low scores for each variable and compare the prediction models). Finally, since the questionnaire was disseminated online, the representativeness of the sample could not be controlled.

## 5. Conclusions

In the present study about adolescents, the relationship between the adoption of protective health behaviors against COVID-19, knowledge about the disease, the perception of the risk involved, tolerance to frustration and planning and decision-making has been confirmed. The development of prevention and communication strategies by the authorities, which take into account the psychosocial factors of adolescents and promote learning about the disease, will contribute to increasing the adoption of protective health behaviors in the context of a pandemic.

## Figures and Tables

**Table 1 jcm-11-04793-t001:** Sociodemographic characteristics, knowledge, attitudes, and risk perceptions, COVID-19 protective health behaviours and psychological variables of the sample.

Variables	Participants (*n* = 354)	Pearson Correlation Coefficient with COVID-19 Protective Health Behaviours	*p*-Value
Age, *years M (SD)*	15.37 (1.55)	−0.089	0.096
Gender			
Male, *n* (%)	119 (33.6)	0.083	0.122
Female, *n* (%)	231 (65.3)		
Other, *n* (%)	4 (1.1)		
Place of residence			
City (more than 10,000 inhabitants), *n* (%)	232 (65.5)	−0.034	0.519
Town (less than 10,000 inhabitants), *n* (%)	122 (34.5)		
What level of education have you completed?			
None, *n* (%)	2 (0.6)	−0.044	0.414
Primary School (6–12 y/o), *n* (%)	163 (46)		
Secondary School (12–16 y/o), *n* (%)	165 (46.6)		
Senior Secondary (16–18 y/o), *n* (%)	20 (5.6)		
DK/NA, *n* (%)	4 (1.1)		
Are any of your relatives you live with currently receiving financial aid?			
Yes, *n* (%)	48 (13.6)	0.084	0.136
No, *n* (%)	267 (75.4)		
DK/NA, *n* (%)	39 (11)		
Knowledge, *M (SD)*	23.45 (2.52)	**0.273**	**<0.001**
Attitudes and risk perceptions, *M (SD)*	38.65 (4.91)	**0.335**	**<0.001**
Family APGAR, *M (SD)*	7.61 (2.32)	**0.137**	**0.010**
Self-efficacy, *M (SD)*	29.20 (5.94)	0.084	0.115
Self-esteem, *M (SD)*	28.07 (6.17)	0.070	0.189
Planning and decision making, *M (SD)*	41.12 (11.02)	**0.175**	**0.001**
Social skills, *M (SD)*	53.86 (10.14)	0.068	0.205
Frustration tolerance, *M (SD)*	26.13 (6.37)	**0.240**	**<0.001**

Note: significant differences (*p* ≤ 0.05) are highlighted in bold. M (SD): mean (standard deviation). DK/NA: Don’t Know/No Answer.

**Table 2 jcm-11-04793-t002:** Linear regression analysis of the influence of knowledge, attitudes, and risk perceptions of COVID-19, and psychological variables on COVID-19 protective health behaviours.

Model	Unstandardised Coefficients		Standardised Coefficients				Collinearity Statistics	
	B	SE	Beta	*t*	*p*-Value	95% CI for B	Tolerance	VIF
(Constant)	**14.355**	4.009		3.581	**<0.001**	[6.471,22.239]		
Attitudes and risk perceptions	**0.405**	0.071	0.280	5.717	**<0.001**	[0.265,0.544]	0.955	1.048
Knowledge	**0.548**	0.138	0.195	3.972	**<0.001**	[0.276,0.819]	0.950	1.052
Frustration tolerance	**0.167**	0.055	0.149	3.028	**0.003**	[0.058,0.275]	0.943	1.061
Planning and decision making	**0.079**	0.031	0.122	2.504	**0.013**	[0.017,0.141]	0.961	1.040

Note: significant differences (*p* ≤ 0.05) are highlighted in bold. SE: standard error. CI: confidence interval. VIF: Variance inflation factor. Dependent Variable: COVID-19 protective health behaviours.

## Data Availability

The data presented in this study are available on request from the corresponding author.

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
