# Peer review of "Associations between Psychological Variables, Knowledge, Attitudes, Risk Perceptions and Health Behaviours towards COVID-19 among Adolescents"

_jcm, 2022, doi:10.3390/jcm11164793_

Round 1
Reviewer 1 Report
Title of the manuscript: The Influence of psychological variables, knowledge and attitude towards COVID-19 on health behaviours among Spanish Adolescents
Dear researcher(s), you are addressing an important and meaningful gap. Your paper is well-written and it has some important results, and if you edit your paper it can be much more effective. Here are some humble suggestions to improve and strengthen the paper. You could increase the effect of your paper with some more recent studies suggested below or any other studies.
Main points:
1. Title:
a. You may consider shorten the title because more and more journals are asking for manuscripts with less number of total words. If the title is brief, comprehensive the readers and researchers will be more likely to benefit from it more. However, you do not have to shorten it.
2. Abstract and keywords: clear and comprehensive
3. Overall language:
- The language is quite clear and well-written. You could use an active language for your future papers throughout the paper since an active language seems to be more effective. And more and more researchers go with an active language. However, you do not have to change for this paper- just a suggestion for your future work and I know some journals asking for a passive language.
4. Length of paragraph : good and
a. you can check the paper and make sure every paragraph is not more than 5 sentences. The best is to stick with 3 to 5 sentences.
5. Introduction: good, but you need to mindful of the following: on page, line 60-62, “However, some results obtained in more recent studies do not corroborate this relationship between self-efficacy and drug use, but find an inverse relationship with self-control and a direct relationship with assertiveness (11).” You need to add additional reference/references, as you state “recent studies” but you only referenced one paper
6. Thoroughness of the literature review: You can support the paper with some recent studies. This will increase the effect of the paper and the journal.
7. Clarity of the description of the Theoretical Framework (TF): Good
8. Research design: Good
9. Clearly providing research questions and/or purpose: Good
10. Choice of research method: very appropriate
11. Appropriateness of procedures chosen for data collection and analysis: well-written
Relevance of data obtained in view of the purpose of the research: good, and you can In Table 1, write the full meaning of ‘’M’’, and ‘’SD’’ as you did for ‘’DK’’ in the footnote
12. Relevance of data obtained in view of the purpose of the research: Good
13. Discussion of the results and their significance: well-written, but you need to be mindful of the following:
-In this statement on page 6, line 228-229 “As for the relationship between knowledge about the disease and the preventive measures with which the population complied,” in this statement you need to be clear about which “disease” I guess is “Covid-19” therefore replace disease with Covid-19
-In this statement on page 6, line 229-230 “previous studies looking at both adults and adolescents confirm this relationship”, here you need to reference this statement
- In this statement on page 6, line 236-237 “in addition to knowledge about the disease,” you need to replace “disease” with Covid-19
- In this statement on page 6, line 246“ Likewise, studies on illness and frustration tolerance in adolescents focus mainly on the study of factors associated with addiction.” You need to reference this
-In this statement on page 6, line 247-248 “Several studies have confirmed that low frustration tolerance is the most common risk factor (52–54).” Clarify risk factor for what pls?
- In this statement on page 6, line 253-254 “previous studies confirm the influence of these variables on the intention to engage in protective health behaviours (56).” You stated previous studies, indicating more than one paper, but you referenced only paper?
- In this statement on page 6, line 257-258, “However, this relationship was not significant in the subsequent regression model with the remaining variables.” You can rephrase this statement to => ‘’However, this relationship was not statistically significant’’
14. Soundness of conclusions in relation to data presented: well-written.
15. Limitation: well-written.
16. Implication:
a. you can increase the effect of your paper by constructing a new section entitled “implication” for clear and brief suggestions in at least two or three of the following most important to you mental health, education, research, administrators, services, etc.: see suggested papers for implications for specific sections
b. I would strongly suggest you to call future researchers to use Online Photovoice (OPV) to conduct research on the same or similar topics. The researchers can use OPV, as one of the most recent and effective innovative qualitative research methods. OPV gives opportunities to the participants to express their own experience with as little manipulation as possible if at all, compared to traditional quantitative methods. As researchers one of our responsibilities is to inform others about recent and effective methods, which will increase the effect of your paper and the journal. Future researchers can conduct only qualitative or mixed method to see if OPV. And educators/trainers etc. also can use OPV for experiential activities to increase group and organizational synergy. Please see suggested papers if you wish to do so.
Armiya’u, A. Y., Yildirim, M., Muhammad, A., Tanhan, A., & Young, J. S. (2022). Mental health facilitators and barriers during covid-19 in Nigeria. Journal of Asian and African Studies. h-tps://doi.org/10.1177/00219096221111354
https://www.researchgate.net/publication/361990976_Mental_Health_Facilitators_and_Barriers_during_Covid-19_in_Nigeria
Doyumğaç, İ., Tanhan, A., & Kıymaz, M. S., (2021). Understanding the most important facilitators and barriers for online education during COVID-19 through online photovoice methodology. International Journal of Higher Education, 10(1), 166-190. https://doi.org/10.5430/ijhe.v10n1p166
https://scholar.google.com/scholar?hl=en&as_sdt=0%2C5&q=Understanding+the+most+important++facilitators+and+barriers+for+online+education+during+COVID-19+through+online+photovoice+methodology&btnG=
Tanhan, A. (2020). Utilizing Online Photovoice (OPV) methodology to address biopsychosocial spiritual economic issues and wellbeing during COVID-19: Adapting OPV to Turkish. Turkish Studies, 15(4), 1029-1086. https://doi.org/10.7827/TurkishStudies.44451
https://scholar.google.com/scholar?hl=en&as_sdt=0%2C5&q=COVID-19+s%C3%BCrecinde+online+seslifoto+%28OSF%29+y%C3%B6ntemiyle+biyopsikososyal++manevi+ve+ekonomik+meseleleri+ve+genel+iyi+olu%C5%9F+d%C3%BCzeyini+ele+almak%3A+OSF%E2%80%99nin+T%C3%BCrk%C3%A7eye+uyarlanmas%C4%B1.+%5BUtilizing+online+photovoice+%28OPV%29+methodology+to+address+biopsychosocial+spiritual+economic+issues+and+wellbeing+during+COVID-19%3A+Adapting+OPV+to+Turkish.%5D&btnG=
17. Figure/tables: Good
18. In-text reference: well written
19. References:
- You could increase the effect of your paper with some more recent studies
- Please use the following link to include all available doi numbers https://doi.crossref.org/simpleTextQuery simply include your reference one or more than one at a time and submit it. Then you should get all doi numbers if a manuscript has it.
In sum, you are addressing an important and meaningful gap. Your paper has some important results, and if you edit your paper based on all or some of the humble suggestions above, it can be much more effective. I have enjoyed reading the paper.
Author Response
Reviewer 1
Open Review
(x) I would not like to sign my review report
( ) I would like to sign my review report
English language and style
( ) Extensive editing of English language and style required
(x) Moderate English changes required
( ) English language and style are fine/minor spell check required
( ) I don't feel qualified to judge about the English language and style
Yes |
Can be improved |
Must be improved |
Not applicable |
|
Does the introduction provide sufficient background and include all relevant references? |
(x) |
( ) |
( ) |
( ) |
Are all the cited references relevant to the research? |
(x) |
( ) |
( ) |
( ) |
Is the research design appropriate? |
(x) |
( ) |
( ) |
( ) |
Are the methods adequately described? |
(x) |
( ) |
( ) |
( ) |
Are the results clearly presented? |
( ) |
(x) |
( ) |
( ) |
Are the conclusions supported by the results? |
(x) |
( ) |
( ) |
( ) |
Comments and Suggestions for Authors
Title of the manuscript: The Influence of psychological variables, knowledge and attitude towards COVID-19 on health behaviours among Spanish Adolescents
Dear researcher(s), you are addressing an important and meaningful gap. Your paper is well-written and it has some important results, and if you edit your paper it can be much more effective. Here are some humble suggestions to improve and strengthen the paper. You could increase the effect of your paper with some more recent studies suggested below or any other studies.
ANSWER: Thank you very much for your thorough review. We are very glad for your comments and feedback. Taking into account your recommendations and those of the rest of the reviewers, we have made quite a few changes in the manuscript.
Main points:
- Title:
- You may consider shorten the title because more and more journals are asking for manuscripts with less number of total words. If the title is brief, comprehensive the readers and researchers will be more likely to benefit from it more. However, you do not have to shorten it.
ANSWER: Thank you. Following this suggestion, the following title is proposed, which we think is the minimum necessary to be able to represent the content of the article: “Associations between psychological variables, knowledge, attitudes, risk perceptions and health behaviours towards COVID-19 among adolescents”.
- Abstract and keywords: clear and comprehensive
ANSWER: Thank you.
- Overall language:
- The language is quite clear and well-written. You could use an active language for your future papers throughout the paper since an active language seems to be more effective. And more and more researchers go with an active language. However, you do not have to change for this paper- just a suggestion for your future work and I know some journals asking for a passive language.
ANSWER: Thank you. We agree. We will keep it in mind for future studies.
- Length of paragraph: good and
- you can check the paper and make sure every paragraph is not more than 5 sentences. The best is to stick with 3 to 5 sentences.
ANSWER: Thank you. We have followed your advised and we have shortened some paragraph.
- Introduction: good, but you need to mindful of the following: on page, line 60-62, “However, some results obtained in more recent studies do not corroborate this relationship between self-efficacy and drug use, but find an inverse relationship with self-control and a direct relationship with assertiveness (11).” You need to add additional reference/references, as you state “recent studies” but you only referenced one paper
ANSWER: Thank you. We have added a reference and rewritten the paragraph: “However, some results obtained in more recent studies do not corroborate this relationship between self-efficacy and drug use, but find that efficacy is a broader construct than typically considered in prevention [11,12]”
- López-Torrecillas, F.; Peralta, I.; Muñoz-Rivas, M.J.; Godoy, J.F. Self-Control and Drug Use. Adicciones 2003, 15, 127–136, doi:10.20882/adicciones.436.
- Choi, H.J.; Krieger, J.L.; Hecht, M.L. Reconceptualizing Efficacy in Substance Use Prevention Research: Refusal Response Efficacy and Drug Resistance Self-Efficacy in Adolescent Substance Use. Health Commun. 2013, 28, 40–52, doi:10.1080/10410236.2012.720245.
- Thoroughness of the literature review:You can support the paper with some recent studies. This will increase the effect of the paper and the journal.
ANSWER: Thank you. Following your suggestion, we have removed some references and added others.
We have removed:
Leal P, Heman A. La baja tolerancia a la frustración y las adicciones. LiberAddictus. 1998;(Marzo):1–5.
Mendoza Sierra MI, Carrasco González AM, Sánchez García M. Consumo de alcohol y autopercepción en los adolescentes españoles. Interv Psicosoc. 2003;12(1):95–111.
We've add:
Choi HJ, Krieger JL, Hecht ML. Reconceptualizing Efficacy in Substance Use Prevention Research: Refusal Response Efficacy and Drug Resistance Self-Efficacy in Adolescent Substance Use. Health Commun [Internet]. 2013 Jan [cited 2022 Jul 25];28(1):40–52. Available from: https://www.tandfonline.com/doi/abs/10.1080/10410236.2012.720245
Fathian-Dastgerdi Z, khoshgoftar M, Tavakoli B, Jaleh M. Factors associated with preventive behaviors of COVID-19 among adolescents: Applying the health belief model. Res Soc Adm Pharm. 2021;
Pell Del Río SM, Santiago DV, Luis A, Rodríguez G, Jesús F, Romero A, et al. Perception during confinement by COVID-19 in a Cuban sample: preliminary results. Acad Ciencias Cuba. 2021;11(1):3–11.
Belintxon M, Calatrava M, Osorio A, Balaguer Á, Vidaurreta M. Internal developmental assets and substance use among Hispanic adolescents. A cross-sectional study. J Adv Nurs. 2022 Jul 1;78(7):1990–2003.
Short, S.E.; Mollborn, S. Social Determinants and Health Behaviors: Conceptual Frames and Empirical Advances. Curr. Opin. Psychol. 2015, 5, 78, doi:10.1016/J.COPSYC.2015.05.002.
Armiya’u, A.Y.; Yıldırım, M.; Muhammad, A.; Tanhan, A.; Young, J.S. Mental Health Facilitators and Barriers during Covid-19 in Nigeria. J Asian Afr Stud 2022, doi:10.1177/00219096221111354.
- Clarity of the description of the Theoretical Framework (TF):Good
ANSWER: Thank you.
- Research design: Good
ANSWER: Thank you.
- Clearly providing research questions and/or purpose: Good
ANSWER: Thank you.
- Choice of research method:very appropriate
ANSWER: Thank you.
- Appropriateness of procedures chosen for data collection and analysis:well-written
Relevance of data obtained in view of the purpose of the research: good, and you can In Table 1, write the full meaning of ‘’M’’, and ‘’SD’’ as you did for ‘’DK’’ in the footnote
ANSWER: Thank you. We have added that information.
- Relevance of data obtained in view of the purpose of the research: Good
ANSWER: Thank you.
- Discussion of the results and their significance:well-written, but you need to be mindful of the following:
-In this statement on page 6, line 228-229 “As for the relationship between knowledge about the disease and the preventive measures with which the population complied,” in this statement you need to be clear about which “disease” I guess is “Covid-19” therefore replace disease with Covid-19
ANSWER: Thank you. We have rewritten it.
-In this statement on page 6, line 229-230 “previous studies looking at both adults and adolescents confirm this relationship”, here you need to reference this statement
ANSWER: Thank you. We have rewritten the paragraph and we have added some new references: “As for the relationship between knowledge about COVID-19 and the preventive measures with which the population complied, previous studies confirm how information focussed on increasing risk perception and knowledge about the disease drive the implementation of protective health behaviours [48,49].”
- Kazaura, M. Knowledge, Attitude and Practices about Dengue Fever among Adults Living in Pwani Region, Tanzania in 2019. Afr. Health Sci. 2020, 20, 1601–1609, doi:10.4314/ahs.v20i4.12.
- Pell Del Río, S.M.; Santiago, D.V.; Luis, A.; Rodríguez, G.; Jesús, F.; Romero, A.; Ruíz, A.L. Perception during Confinement by COVID-19 in a Cuban Sample: Preliminary Results. Acad. Ciencias Cuba 2021, 11, 3–11.
- In this statement on page 6, line 236-237 “in addition to knowledge about the disease,” you need to replace “disease” with Covid-19
ANSWER: Thank you. We have rewritten it.
- In this statement on page 6, line 246“ Likewise, studies on illness and frustration tolerance in adolescents focus mainly on the study of factors associated with addiction.” You need to reference this
-In this statement on page 6, line 247-248 “Several studies have confirmed that low frustration tolerance is the most common risk factor (52–54).” Clarify risk factor for what pls?
ANSWER: Thank you. Taking both suggestion into account, we have rewritten the following paragraph:
“Likewise, studies on illness and frustration tolerance in adolescents focus mainly on the study of factors associated with addictive risk. They have confirmed that low frustration tolerance is the most common risk factor associated with addiction [55–57]”
- Ávila Fray, D.X.; Pilco Guadalupe, A. Tolerancia a La Frustración y Distorsiones Cognitivas En Estudiantes Con Consumo de Alcohol. Rev. Eugenio Espejo 2016, 10, 13–22, doi:10.37135/ee.004.01.01.
- Díaz, C.A.G.; Arévalo, J.B.; Angarita, E.V.; Ruiz, Y.S. Relación Entre El Consumo Excesivo de Alcohol y Esquemas Maladaptativos Tempranos En Estudiantes Universitarios. Rev. Colomb. Psiquiatr. 2010, 39, 362–374, doi:10.1016/s0034-7450(14)60256-0.
- Dorta, Y.O.; Hernández, I.O.; Vento, E.M.; Adam, M.R.S.; Martínez, E.M.Á. Factores Individuales de Riesgo Adictivo En Estudiantes de Noveno Grado. Nueva Paz, 2015. Rev. del Hosp. Psiquiatr. la Habana 2015, 13.
- In this statement on page 6, line 253-254 “previous studies confirm the influence of these variables on the intention to engage in protective health behaviours (56).” You stated previous studies, indicating more than one paper, but you referenced only paper?
ANSWER: Thank you. We have added the following reference:
Belintxon M, Calatrava M, Osorio A, Balaguer Á, Vidaurreta M. Internal developmental assets and substance use among Hispanic adolescents. A cross-sectional study. J Adv Nurs. 2022 Jul 1;78(7):1990–2003.
- In this statement on page 6, line 257-258, “However, this relationship was not significant in the subsequent regression model with the remaining variables.” You can rephrase this statement to => ‘’However, this relationship was not statistically significant’’
ANSWER: Thank you. We have rewritten it.
- Soundness of conclusions in relation to data presented:well-written.
ANSWER: Thank you.
- Limitation: well-written.
ANSWER: Thank you.
- Implication:
- you can increase the effect of your paper by constructing a new section entitled “implication” for clear and brief suggestions in at least two or three of the following most important to you mental health, education, research, administrators, services, etc.: see suggested papers for implications for specific sections
ANSWER: Thank you. As we have observed that this joural does not usually have its own section on implications, what we have done is add them in conclusions:
“In the present study about adolescents, the relationship between the adoption of protective health behaviors against COVID-19, knowledge about the disease, the percep-tion of the risk involved, tolerance to frustration and planning and decision-making has been confirmed. The development of prevention and communication strategies by the au-thorities, which take into account the psychosocial factors of adolescents and promote learning about the disease, will contribute to increasing the adoption of protective health behaviors in the context of a pandemic.”
- I would strongly suggest you call future researchers to use Online Photovoice (OPV) to conduct research on the same or similar topics. The researchers can use OPV, as one of the most recent and effective innovative qualitative research methods. OPV gives opportunities to the participants to express their own experience with as little manipulation as possible if at all, compared to traditional quantitative methods. As researchers one of our responsibilities is to inform others about recent and effective methods, which will increase the effect of your paper and the journal. Future researchers can conduct only qualitative or mixed method to see if OPV. And educators/trainers etc. also can use OPV for experiential activities to increase group and organizational synergy. Please see suggested papers if you wish to do so.
ANSWER: Thank you for that information. The mention of the use of Online Photovoice (OPV) has been added in limitations, as a possible method of improving the study. We have also cited the paper that seemed most related to our study, so that it can be taken as an example of the use of the OPV: “It would also be interesting to analyze the facilitators and barriers that influence the modi-fication of protective health behaviors of adolescents, through qualitative methods such as Online Photovoice (OPV), which gives opportunities to the participants to express their own experience with the least amount of manipulation possible [77].”
Armiya’u, A.Y.; Yıldırım, M.; Muhammad, A.; Tanhan, A.; Young, J.S. Mental Health Facilitators and Barriers during Covid-19 in Nigeria. J Asian Afr Stud 2022, doi:10.1177/00219096221111354.
- Figure/tables: Good
ANSWER: Thank you.
- In-text reference:well written
ANSWER: Thank you.
- References:
- You could increase the effect of your paper with some more recent studies
- Please use the following link to include all available doi numbers https://doi.crossref.org/simpleTextQuery simply include your reference one or more than one at a time and submit it. Then you should get all doi numbers if a manuscript has it.
ANSWER: Thank you so much for that resource. We have added all the doi and some new references following your suggestions.
In sum, you are addressing an important and meaningful gap. Your paper has some important results, and if you edit your paper based on all or some of the humble suggestions above, it can be much more effective. I have enjoyed reading the paper.
ANSWER: Thank you so much for your review and your words. We believe that the manuscript has improved remarkably thanks to your work.
Reviewer 2 Report
This paper describes a study among Spanish adolescents on psychosocial variables influencing “practices related to COVID-19 /// adherence to preventive measures”
Although technically, there is nothing wrong with this paper or the study, there are several aspects that could be improved: the data seem interesting but the analyses are too general while detailed reporting is certainly possible with the data that the authors have.
- - Next to general correlations between general scales, there should be attention to the specific items.
- - it took me some effort to find out where the measures of these “health behaviors” were reported; that should be described more clearly including the outcomes of those measures.
- - I would also suggest that the authors provide access to the (English) questionnaire in an additional file.
- - The data provide the possibility of comparing scores, between high-low scores groups for behavior, on the items for the predictor variables. That would at least provide more information for future health education. Moreover, not all preventive behavior are the same in terms of compliance; it would be interesting to see more detail including the correlations with the predictors.
Author Response
Reviewer 2
Open Review
( ) I would not like to sign my review report
(x) I would like to sign my review report
English language and style
( ) Extensive editing of English language and style required
( ) Moderate English changes required
(x) English language and style are fine/minor spell check required
( ) I don't feel qualified to judge about the English language and style
Yes |
Can be improved |
Must be improved |
Not applicable |
|
Does the introduction provide sufficient background and include all relevant references? |
( ) |
(x) |
( ) |
( ) |
Are all the cited references relevant to the research? |
(x) |
( ) |
( ) |
( ) |
Is the research design appropriate? |
(x) |
( ) |
( ) |
( ) |
Are the methods adequately described? |
( ) |
( ) |
(x) |
( ) |
Are the results clearly presented? |
( ) |
(x) |
( ) |
( ) |
Are the conclusions supported by the results? |
( ) |
(x) |
( ) |
( ) |
Comments and Suggestions for Authors
This paper describes a study among Spanish adolescents on psychosocial variables influencing “practices related to COVID-19 /// adherence to preventive measures”
Although technically, there is nothing wrong with this paper or the study, there are several aspects that could be improved: the data seem interesting but the analyses are too general while detailed reporting is certainly possible with the data that the authors have.
ANSWER: Thank you very much for your thorough review. We are very glad for your comments and feedback. Taking into account your recommendations and those of the rest of the reviewers, we have made quite a few changes in the manuscript.
- - Next to general correlations between general scales, there should be attention to the specific items.
ANSWER: Thank you. That item-by-item analysis could be interesting. We have worked with the total score of the scale since they are validated instruments with their total score. In addition, as we used many instruments in this study, it would take too long to discuss item by item and deviate from the main objective, which is to work with the general construct.
- - it took me some effort to find out where the measures of these “health behaviors” were reported; that should be described more clearly including the outcomes of those measures.
ANSWER: Thank you. We have added the definition and explanation of them in the introduction “Health behaviours are practises made by persons that impact health or mortality and can either benefit or harm the actor's or others' health [19].”
- Short, S.E.; Mollborn, S. Social Determinants and Health Behaviors: Conceptual Frames and Empirical Advances. Curr. Opin. Psychol. 2015, 5, 78, doi:10.1016/J.COPSYC.2015.05.002.
and
“In terms of protective health behaviours related to COVID-19 (compliance with the preventive measures set by the health authorities, e.g.: safety distance, hand washing, wearing a mask) or risk behaviours (non-compliance with these measures) […]”
In addition, we have unified the different terms on health behaviors to “protective health behaviours”.
- - I would also suggest that the authors provide access to the (English) questionnaire in an additional file.
ANSWER: Thank you. The original questionnaire in English can be found in this cited article:
- Honarvar, B.; Lankarani, K.B.; Kharmandar, A.; Shaygani, F.; Zahedroozgar, M.; Rahmanian Haghighi, M.R.; Ghahramani, S.; Honarvar, H.; Daryabadi, M.M.; Salavati, Z.; et al. Knowledge, Attitudes, Risk Perceptions, and Practices of Adults toward COVID-19: A Population and Field-Based Study from Iran. Int. J. Public Health 2020, 65, 731–739, doi:10.1007/s00038-020-01406-2.
What our team did in a previous study also cited, was to validate that scale in Spanish:
- Aguilar-Latorre, A.; Asensio-Martínez, Á.; García-Sanz, O.; Oliván-Blázquez, B. Knowledge, Attitudes, Risk Perceptions, and Practices of Spanish Adolescents Toward the COVID-19 Pandemic: Validation and Results of the Spanish Version of the Questionnaire. Front. Psychol. 2022, 12, 1–9, doi:10.3389/fpsyg.2021.804531.
- - The data provide the possibility of comparing scores, between high-low scores groups for behavior, on the items for the predictor variables. That would at least provide more information for future health education. Moreover, not all preventive behavior are the same in terms of compliance; it would be interesting to see more detail including the correlations with the predictors.
ANSWER: Thank you. It is a very good idea that we were thinking about. For doing this, we need to increase the sample size, so that when doing subgroup analysis, the different categories are compensated and similar. So, we've added this in the limitations part: “Furthermore, future studies with larger sample size could perform subgroup analyzes (i.e., divide the sample according to whether they have high or low scores for each variable and compare the prediction models).”
Round 2
Reviewer 2 Report
the authors responded well to my feedback.